# The Efficacy of Workplace Interventions on Improving the Dietary, Physical Activity and Sleep Behaviours of School and Childcare Staff: A Systematic Review

**DOI:** 10.3390/ijerph17144998

**Published:** 2020-07-11

**Authors:** Nicole Nathan, Beatrice Murawski, Kirsty Hope, Sarah Young, Rachel Sutherland, Rebecca Hodder, Debbie Booth, Elaine Toomey, Sze Lin Yoong, Kathryn Reilly, Flora Tzelepis, Natalie Taylor, Luke Wolfenden

**Affiliations:** 1Hunter New England Population Health, Hunter New England Area Health Service, Longworth Avenue, Wallsend, NSW 2287, Australia; sarah.young6@health.nsw.gov.au (S.Y.); rachel.sutherland@health.nsw.gov.au (R.S.); rebecca.hodder@health.nsw.gov.au (R.H.); serene.yoong@health.nsw.gov.au (S.L.Y.); kathryn.reilly@health.nsw.gov.au (K.R.); flora.tzelepis@health.nsw.gov.au (F.T.); luke.wolfenden@health.nsw.gov.au (L.W.); 2School of Medicine and Public Health, The University of Newcastle, University Drive, Callaghan, NSW 2308, Australia; beatrice.murawski@health.nsw.gov.au (B.M.); kirsty.hope@health.nsw.gov.au (K.H.); 3Priority Research Centre for Health Behaviour, The University of Newcastle, University Drive, Callaghan, NSW 2308, Australia; 4Hunter Medical Research Institute, Kookaburra Circuit, New Lambton Heights, NSW 2305, Australia; 5University Library, Academic Division, University of Newcastle, University Drive, Callaghan, NSW 2308, Australia; debbie.booth@newcastle.edu.au; 6Health Behaviour Change Research Group, School of Psychology, National University of Ireland, University Road, Galway H91 TK33, Ireland; elaine.toomey@nuigalway.ie; 7Cancer Research Division, Cancer Council New South Wales, 153 Dowling St, Woolloomooloo, NSW 2011, Australia; natalie.taylor@nswcc.org.au; 8School of Health Sciences, University of Sydney, Camperdown, NSW 2006, Australia

**Keywords:** workplace, health promotion, school staff, physical activity, diet, sleep health

## Abstract

There is a need for effective interventions that improve the health and wellbeing of school and childcare staff. This review examined the efficacy of workplace interventions to improve the dietary, physical activity and/or sleep behaviours of school and childcare staff. A secondary aim of the review was to assess changes in staff physical/mental health, productivity, and students’ health behaviours. Nine databases were searched for controlled trials including randomised and non-randomised controlled trials and quasi-experimental trials published in English up to October 2019. PRISMA guidelines informed screening and study selection procedures. Data were not suitable for quantitative pooling. Of 12,396 records screened, seven articles (based on six studies) were included. Most studies used multi-component interventions including educational resources, work-based wellness committees and planned group practice (e.g., walking groups). Multiple outcomes were assessed, findings were mixed and on average, there was moderate risk of bias. Between-group differences in dietary and physical activity behaviours (i.e., fruit/vegetable intake, leisure-time physical activity) favoured intervention groups, but were statistically non-significant for most outcomes. Some of the studies also showed differences favouring controls (i.e., nutrient intake, fatty food consumption). Additional robust studies testing the efficacy of workplace interventions to improve the health of educational staff are needed.

## 1. Introduction

The World Health Organization (WHO) has identified the workplace as an important setting to implement population-wide health promotion interventions [1]. With more than 60 million staff globally [2], educational settings, including childcare centres, elementary and secondary schools, have the potential to deliver wide-reaching behaviour change interventions. Moreover, the equipment and facilities that are already available in these settings (e.g., canteens, sports grounds), combined with regular opportunities to engage in healthy behaviours (e.g., active participation during supervision or sports lessons) further supports the long-term implementation of health promoting strategies. Thus, interventions implemented in this setting have the potential to significantly influence the health behaviours of a large number of adults [3]. Educational staff may benefit from effective behaviour change interventions, as the evidence suggests that they report high levels of obesity, poor diet, insufficient physical activity and poor sleep health [4,5,6,7,8]. This may be due to high levels of stress and time constrictions [9], leaving little opportunity or motivation to eat more healthily and engage in sufficient physical activity, especially at moderate to vigorous intensity. Improvements in one or more of these health behaviours in educational staff may also have a broader impact by significantly reducing the costs associated with staff absenteeism due to lifestyle-related medical conditions [10,11,12,13], as well as costs associated with presenteeism (i.e., impaired functioning at work due to a medical condition) [14]. Finally, it has been hypothesised that improving staff health in educational settings may have a positive “spill-over” effect on students’ health behaviours given teachers’ influence as role models to children [15]. That is, teachers who consistently engage in healthy dietary and physical activity practices can convey a positive message to students promoting a healthy lifestyle [16]. Moreover, educational staff who follow a healthy lifestyle may be more likely to implement policies and practices specific to student health [17], whilst driving the formation of an overall social environment that is conducive of healthy behaviours [18].

Previous reviews and meta-analyses of the efficacy of workplace interventions conducted in other target groups such as office workers or factory staff have demonstrated mixed to modest effects on adults’ dietary, physical activity and sleep behaviours [19,20,21,22,23,24,25]. However, none of these syntheses have focused exclusively on educational settings. Therefore, a synthesis of available evidence targeting the health behaviours of the educational workforce is needed to determine if improvements in behavioural outcomes can be achieved through health promoting interventions in educational settings.

The primary aim of the current review was to assess the efficacy of workplace interventions targeting school or childcare staff to improve their dietary, physical activity and/or sleep behaviours. A secondary aim of the review was to assess if such interventions influenced staff physical health (e.g., body mass index (BMI)), mental health (e.g., stress, anxiety), workplace productivity (e.g., absenteeism), or students’ health behaviours, as these outcomes are known to be associated with the health behaviours of interest (i.e., diet, physical activity, sleep).

## 2. Materials and Methods

### 2.1. Registration

The review was prospectively registered with the international prospective register of systematic reviews (PROSPERO; CRD42018107750) and adhered to the preferred reporting items for systematic reviews and meta-analyses (PRISMA) guidelines [26].

### 2.2. Eligibility Criteria

This review included studies published in English and indexed between database inception and 31 October 2019.

#### 2.2.1. Study Designs

Eligible study designs included any study that had a parallel comparison, such as individual-randomised controlled trials (I-RCTs), cluster-randomised controlled trials (C-RCTs) non-randomised controlled clinical trials, or quasi-experimental designs. Eligible comparison groups were usual care, waitlist-control, minimal or an alternate intervention.

#### 2.2.2. Participants

Studies were eligible for inclusion if participants were staff employed at elementary or secondary schools or childcare centres, including teachers, childcare workers, school principals, managerial and administrative staff, or other staff directly affiliated with the institution/educational facility (casual, part-time and full-time staff were eligible).

#### 2.2.3. Interventions

This review included any educational, experiential and/or health promoting workplace intervention to change the dietary, physical activity and/or sleep behaviours of school or childcare staff, delivered via the educational setting. Interventions that targeted single behaviours (i.e., physical activity only) or multiple behaviours (i.e., nutrition and physical activity in combination) were included. Studies were also included if they primarily targeted student health behaviours (e.g., classroom physical activity), but actively involved staff in the intervention, or engaged staff to deliver the intervention to students and also measured outcomes in school and/or childcare staff. Studies were excluded if interventions were treating doctor-diagnosed diseases, including eating disorders (e.g., anorexia nervosa, bulimia), obesity, or clinical sleep disorders (e.g., chronic insomnia, sleep apnoea), as the types of interventions needed to treat these conditions are likely to be very specific and require support from clinical practitioners.

#### 2.2.4. Outcomes

Eligible studies had to report at least one of the following primary behavioural outcomes: dietary behaviours (e.g., fruit, vegetable, sweetened beverages, water, total energy intake) measured via direct observation, diary entries, surveys or questionnaires, or food purchasing history, and any other forms of assessment (e.g., technology-based food intake measurement via camera); physical activity (e.g., minutes of moderate- to vigorous-intensity physical activity (MVPA), daily step counts, number of days or frequency of resistance training (RT)) measured using devices (e.g., via accelerometer or pedometer) and/or via self-report (e.g., diaries or surveys); or sleep (e.g., parameters of sleep health such as sleep quality, sleep duration, sleep/wake timing, etc.) measured using device-measured (e.g., accelerometers, home-based polysomnography) or subjective methods (e.g., questionnaires assessing sleep quality, sleep duration, daytime complaints, or severity of sleep problems).

Secondary outcomes may have included physical health including measures of adiposity (e.g., BMI, waist-to-hip ratio) assessed using device-measured methods (e.g., stadiometer, DEXA, maximal oxygen intake (VO_2_max)) or subjective methods (e.g., self-reported height and weight used to calculate BMI); mental health (e.g., depression, anxiety, stress); workplace productivity (e.g., job performance, absenteeism/presenteeism, job satisfaction); students’ health behaviour such as dietary intake, physical activity levels or sleep evaluated using the same device-measured and/or subjective measures as listed above for primary outcomes.

### 2.3. Information Sources and Search Strategy

A comprehensive search strategy using adaptations of previously published search strings was developed in consultation with an academic librarian (DB) [27,28]. To identify original articles and grey literature (e.g., research reports, policy documents, conference proceedings), database searches were conducted in: CINAHL, ProQuest Dissertation and Theses, EBSCO Megafile Ultimate, EMBASE, ERIC, MEDLINE, PsycINFO, Scopus and SportDiscus. Search strategies were developed in MEDLINE and adapted according to each individual database (see Appendix A). Term sets were specific to the settings of interest (e.g., schools, childcare), the occupation (e.g., teacher, principal, director), and the target health behaviours (i.e., diet, physical activity and sleep). Additionally, clinical trial registry searches of the WHO international clinical trials registry platform and the U.S. National Institutes of Health database were conducted using relevant terms. To identify additional articles, the reference lists of all included studies were screened, and authors were contacted if no published full-text article was available.

### 2.4. Study Selection

Double independent screening of the titles and abstracts of identified records was conducted by multiple authors (N.N., B.M., K.H., R.H., K.R., E.T.). Full-text articles of potentially relevant studies were obtained and independently assessed against inclusion criteria by the authors in teams of two (N.N., B.M., K.H., S.Y.). Disagreement regarding the eligibility of a study was resolved by discussion and consensus. The number of articles at each screening stage is shown in Figure 1.

### 2.5. Data Collection Process

Extraction of relevant information was completed by two reviewers (B.M., S.Y.) using an adapted version of the Cochrane data collection form for reviews of intervention studies (RCTs and non-RCTs) [29]. A third reviewer (N.N.) checked extracted data for accuracy. Discrepancies in extracted data were resolved through discussion and consensus. The following information was extracted: study aim, setting, country, study design, number randomised or allocated to intervention groups (for non-randomised trials), intervention components, intervention duration (i.e., number of weeks, months) and theoretical framework underpinning the intervention, the individual study’s primary and secondary study outcomes, measures and results (mean and standard deviation (SD) data for all continuous outcomes) and any information required to assess risk of bias (e.g., participant allocation). Study authors were contacted to obtain additional information if any key data relevant for synthesis were missing.

### 2.6. Risk of Bias

Risk of bias was assessed independently by two reviewers (NN, BM) using two different tools to evaluate studies that reported findings from (1) cluster-randomised trials and (2) non-randomised trials. Where needed, this was discussed with a third reviewer (RH). Scoring protocols observed the published instructional material for both of the two tools used to evaluate risk of bias [30,31].

Using the Cochrane Tool for assessing risk of bias (including extended criteria for the assessment of cluster randomised controlled trials (C-RCTs)) [30], each cluster-RCT was assessed as being at ‘high’, ‘low’ or ‘unclear’ risk of bias for: random sequence generation, allocation concealment, blinding of participants and researchers, blinding of outcome assessment, incomplete outcome data and selective outcome reporting [30]. Extended criteria (for C-RCTs) included recruitment bias, loss of clusters, analysis, contamination and baseline discrepancies.

Using an adapted version of the Newcastle Ottawa Quality Assessment Scale [31], non-randomised trials received points in three main categories (i.e., participant/sample selection, comparability and outcome) across a total of seven items. Item 4 of the selection category from the original scale (i.e., demonstration that outcome of interest was not present at start of study) was omitted due to limited applicability in the context of this review; and item 2 of the outcome category (i.e., follow-up long enough for outcomes to occur) was amended from 6 months to 3 months, as this was deemed sufficient for changes in diet or PA to occur [32,33].

### 2.7. Data Synthesis

Trial heterogeneity was described by specifying participant, intervention, comparison and outcome characteristics of included studies. The high level of heterogeneity observed in the included trials precluded meta-analysis. Therefore, results were narratively synthesised by reporting the effect of interventions by outcome measure. For trials with multiple follow-up periods, data from the first post-intervention follow-up assessment was prioritised to describe intervention effects, since the objective of the current review was to summarise evidence on the efficacy of interventions, rather than their sustainability. Adjusted results were used if a study reported results from both, unadjusted and adjusted analyses. Behavioural outcomes (i.e., changes in physical activity, dietary and/or sleep behaviour) were of main interest to the narrative synthesis; however, where effects for the specified secondary outcomes were reported these were also described narratively.

## 3. Results

### 3.1. Study Selection

A total of 12,396 titles and abstracts were screened for eligibility, of which 12,271 were excluded. Of the remaining 125 articles included for full-text assessment, 118 were excluded as they did not meet eligibility criteria (see Figure 1). Seven articles [17,34,35,36,37,38,39] reporting on six unique studies were included in this review. No additional articles were obtained through author contact.

### 3.2. Study Characteristics

#### 3.2.1. Study Designs

The characteristics of the included studies are summarised in Table 1.

Of the six included studies, two were conducted in the U.S. [17,37], and one trial each in Canada [36], China [38], South Africa [39] and Taiwan [34]. Three of the studies employed cluster randomised-controlled trial designs [17,37,38], and three were conducted using quasi-experimental designs [34,36,39]. Trials were conducted between 1996 [36] and 2015 [39] and had a duration between 4 months [36] and 6 months [38] and up to 2 years [17,34,37].

#### 3.2.2. Participants

All of the six included studies were conducted in school settings. No studies were conducted in childcare centres (see Table 1). Five were conducted in elementary schools, with the number of participating schools ranging from two [39] to 32 [17]. One study was conducted in two middle schools [38]. Sample sizes ranged from 23 [39] to 364 [34] teachers and the average age of participants ranged from 20 [34] to 54 years [36]. All of the included studies reported a higher proportion (up to 98% [17]) of female participants than males. Four studies included teaching staff only [17,34,38,39], and two targeted all school staff (e.g., including principals, secretaries) [36,37]. Two studies also reported participant characteristics at the student level [17,39]. One of these studies [39] included data from 681 fifth Grade (mean age = 10.5 (SD = 1.2) years) and sixth Grade students (mean age = 11.6 (SD = 1.0). The other study [17] included data from 2708 third through fifth Grade students with an average age of 8.7 years, 53% of whom were female.

#### 3.2.3. Interventions

The overarching foci of the interventions included obesity and cardiovascular disease prevention and health promotion by way of behaviour change. Three of the trials aimed to improve school staff’s physical activity and dietary behaviours in combination [17,36,37], and the other three trials targeted dietary behaviours only [34,38,39]. None have targeted sleep health.

A detailed overview of intervention characteristics (e.g., use of theory, mode of delivery) based on TIDIER checklist items is provided as a Appendix A. Five of the trials [17,36,37,38,39] utilised multi-component interventions (e.g., educational workshop combined with materials, organised exercise programs or personalised advice based on health checks); however, no two trials used the same combination of intervention components. One of the studies [34] provided insufficient detail on the intervention components that were used as part of implementing the health-promoting schools framework.

The intervention components most commonly reported were provision of educational materials [17,36,38,39], formation of workplace wellness committees or groups [37,38], and organised activities such as walking or aerobic classes [17,37]. Studies also distributed promotional materials (e.g., posters) [38,39], and gave out incentives for participation (e.g., cash stipends, t-shirts, grocery shop gift certificates) [17,37]. Four studies reported using theoretical or conceptual frameworks to design the intervention, including Social Cognitive Theory [37,39], the Meaningful Learning Model [39], and the Health Promoting Schools Framework [34,38]. However, no details were reported to demonstrate how theory was operationalised. Another study also stated that the student-focussed program (i.e., Gimme-5) that was administered to control schools was based on Social Cognitive Theory, while none of the components that formed the intervention were reported as being theory-based [17].

#### 3.2.4. Comparator Conditions

Intervention conditions were compared against no intervention control groups in four studies [36,37,38,39] and another study offered control group participants the Gimme-5 program (i.e., a health education curriculum to increase students’ consumption of fruit and vegetables), which was part of a more comprehensive program delivered to the intervention group [17]. One of the trials had two active comparator conditions and offered a non-diet version of the health promoting schools program to one of the control groups, whereas the other group served as non-health-promoting schools (i.e., usual practice) [34].

#### 3.2.5. Outcomes

Primary outcomes. Studies that targeted dietary behaviours of school staff assessed intervention effects by way of measuring consumption of fruits and vegetables [17,34,37], dairy [38], discretionary foods [34,38], sugar-sweetened beverages (e.g., soft drinks) [34,38], water [34], and fatty foods [17,34,36,38] dietary practices (personal, at school, in the classroom, and schoolwide) [39], and whether participants were eating breakfast [34,38]. The three studies targeting physical activity examined intervention effects by assessing changes in staff’s weekly moderate- to vigorous-intensity physical activity (MVPA), reported as metabolic equivalents of task (METs) [17,37], total weekly minutes and total energy expenditure (kcal) [17] and as an exercise behaviour score [36].

Secondary outcomes. The physical health outcomes assessed in the included studies were changes in blood pressure [17], sum of skinfolds [17], body mass index (BMI) [37] and waist-to-hip ratio [17,37]. None of the included studies assessed participants’ mental health outcomes. One study reported measuring job satisfaction and organisational climate as intervention outcomes [17]. Only one of the included studies examined a number of behavioural outcomes at the student level (i.e., fruit and vegetable preferences, nutrition knowledge and dietary intake) [17].

Follow-up data (first follow-up post-intervention) were collected between 4 months [36] and 2 years after baseline [17,34,37]. One study conducted an additional long-term follow up at 3 years [17] and the remaining study did not report sufficient detail to determine when follow up data collection occurred [39].

#### 3.2.6. Retention and Adherence

Three of the four studies with a single follow-up assessment reported participant retention rates that ranged from 30.3% [37] to 100% [38]. One study did not report retention rates [34] and the study that conducted two follow-up assessments reported a retention rate of 41% at years two and three [17]. One study assessed program participation and reported a 21% attendance rate for health classes [17], whereas up to one fourth of the lectures were attended by 72% of teachers. Another study reported that 70% of intervention group participants participated in the health promotion activities that were offered [37]. One of these studies examined the impact of program participation (i.e., attendance) on intervention efficacy, but found no difference between low and high attendance rates [17].

### 3.3. Risk of Bias

#### 3.3.1. Study Quality of the Included Cluster-Randomised Controlled Trials

Of the three cluster-randomised controlled trials [17,37,38], none reported how the random allocation sequences were generated and whether allocation was concealed (see Table 2 for ratings). Therefore, it was unclear to what extent these studies had a selection bias. While it was not possible to blind participants to the groups they were allocated to (due to the nature of the interventions), none of the studies reported blinding of researchers or outcome assessors. Therefore, performance bias was deemed high across these studies and detection bias was deemed unclear. Two of the studies had a low attrition bias, due to having suffered no loss-to-follow up [38], or because results from complete cases as well as all cases were reported [17], whereas another study reported list-wise deletion of missing cases after substantial loss to follow up and therefore had a high risk of attrition bias [37]. The risk of bias associated with the statistical analyses used was low in all three of the cluster-randomised controlled trials. Since no study protocols or trial registry records were available for any of the studies to confirm this, reporting bias was deemed unclear. Risk of other biases was high (due to loss of clusters) in one study [17], unclear (due to unclear recruitment bias and unclear loss of clusters) in another [37] and low in the third study [38].

#### 3.3.2. Study Quality of the Included Non-Randomised Controlled Trials

Overall study quality in all three of the non-randomised controlled trials [34,36,39] was deemed moderate. One study received an overall rating of 6 out of a possible 8 (for representativeness of the cohort, selection of the non-intervention cohort from the same community as the intervention cohort, ascertainment of intervention, comparability of the cohorts (double scores) and sufficient follow-up time for outcomes to occur) [36]. Another study received an overall rating of 5 (for selection of the non-intervention cohort, ascertainment of intervention, comparability of cohorts (single score), sufficient follow-up time, and adequacy of follow-up of cohorts) [39]. The third study received an overall rating of 5 (for representativeness of the cohort, selection of the non-intervention cohort from the same community as the intervention cohort, comparability of cohorts (double scores) and sufficient follow-up time) [34]. Further detail is provided in Table 3.

### 3.4. Intervention Efficacy

A summary of findings for each of the included studies is provided in Table 1. As the studies were heterogeneous in program duration, intervention content, reported theoretical underpinnings and outcome measures, they could not be combined in a quantitative synthesis.

#### 3.4.1. Intervention Effects on Diet

In one of the studies that targeted diet using the Health Promoting Schools Framework, multivariate linear regression models adjusted for gender, age, marital status, years of teaching, teaching status and being a health education teacher, showed a statistically significant between-group difference for consumption of fruit and vegetables (*β* = 1.0, 95% CI: 0.2, 1.8, *p* = 0.01) in favour of the dietary intervention group (relative to the no intervention control (NHP) group) [34]. In this study, intervention group participants also had lower nutrient intake behaviour scores (*β* = −0.6, 95% CI: −1.8, 0.6) and reported higher fatty food consumption (*β* = 0.3, 95% CI: −0.4, 0.9) than those in the control group (NHP); however, differences between groups were statistically non-significant [34]. The second study that employed a Health Promoting Schools program found greater adoption rates for healthy eating behaviours (i.e., consumption of fresh fruits and vegetables, dairy products, breakfast, dessert, fried foods and soft drinks) in those who received the intervention, compared to controls [38]. Differences between groups were statistically non-significant for all outcomes in this study (NB. no statistics reported) [38]. In another study, teachers’ who received a nutrition education intervention reported greater improvements in their dietary practices (i.e., personal dietary practices, dietary practices at school, classroom and schoolwide food practices, and practices in food hygiene) than teachers who did not receive the intervention [39]. Differences between groups, however, were not statistically significant in this study [39].

The multi-component health screening and counselling program that targeted diet and physical activity in combination found an unexpected increase in fat consumption scores that was greater in the intervention group [36]. Relative to control group participants, however, the difference in change scores, adjusted for baseline values of the outcome and potential confounders, was not statistically significant (*p* = 0.15) [36]. The findings from the Johnson and Johnson Live for Life program were similar in that no statistically significant differences were detected between groups for dietary practices (i.e., fruit and vegetable intake) [17]. Reporting on a workplace obesity prevention program targeting U.S. school staff, the authors of the study found that participants in the intervention group reported greater improvements in daily fruit and vegetable intake than controls [37]. Between-group differences in this trial, however, were of negligible magnitude and statistically non-significant (*p* = 0.619) [37].

#### 3.4.2. Interventions Effects on Physical Activity

One of the studies that sought to improve diet and physical activity in combination reported that after 4 months, a significantly greater proportion of participants in the intervention group increased their leisure-time physical activity levels, relative to participants in the comparison group (62.1% versus 47.3%; *p* = 0.02) [36]. However, the adjusted mean difference in exercise behaviour scores (i.e., 4.6 in the intervention group versus −0.4 in the control group) was not statistically significant (*p* = 0.05) [36]. No statistically significant between-group differences were found in physical activity in the remaining two studies [17,37], one of which reported a small increase (+2.32 min) in the control group and a slight decrease (−0.36 min) in physical activity in the intervention group [37].

#### 3.4.3. Intervention Effects on Physical Health

The obesity prevention study assessed between-group differences in changes in participants’ BMI (analyses adjusted for age, ethnicity, job classification and school clusters) and found a statistically significant effect in favour of the intervention group with a mean reduction of 0.04 kg/m^2^ in the intervention group, compared to a 0.37 kg/m^2^ increase in controls (equals a difference of 0.41 difference, *p* = 0.48) [37]. This study reported unchanged measurements for participants’ waist-to-hip ratios in both groups after two years. The second study that assessed participants’ anthropometric outcomes (i.e., waist-to-hip ratio, blood pressure, sum of skinfolds) also reported no statistically significant intervention effects [17].

#### 3.4.4. Intervention Effects on Workplace Productivity

The study that examined teachers’ job satisfaction and perceived organisational climate reported that no statistically significant group differences were found for these outcomes [17].

#### 3.4.5. Intervention Effects on Student Health Behaviours

One study assessed the impact of the intervention on student outcomes, and results based on complete case data showed that, in line with what was observed at the staff level, the intervention delivered to teachers had no statistically significant spill-over effect on student outcomes (i.e., anthropometric measures, dietary intake, fruit and vegetable knowledge and preferences) [17]. Using incomplete data (i.e., full sample), however, there was a statistically significant effect on students’ fruit and vegetable preferences (*p* < 0.01) favouring the control group (*p* < 0.001), which had only received the Gimme-5 program [17]. The health screening study that successfully targeted dietary and physical activity behaviours in Canadian elementary school teachers did not assess behavioural outcomes at the student level but examined the frequency at which teachers discussed heart health with students [36]. Change scores for this outcome differed significantly between groups at the follow-up in favour of the intervention [36]. A third study showed statistically significant between-group differences in students’ nutrition knowledge (*p* = 0.001) and attitudes (*p* = 0.002) in favour of the intervention group [39]. However, no between-group differences were found for students’ dietary practices, which had decreased (i.e., worsened) in both groups following the intervention [39]. This was in contrast to positive (albeit statistically non-significant) changes at the staff level [39].

## 4. Discussion

The teaching workforce has been identified as a relatively novel target group for the implementation of workplace-based health promotion initiatives. To our knowledge, this review contributes new knowledge on the efficacy of workplace interventions targeting the dietary, physical activity and/or sleep behaviours of school and childcare staff. Given the limited number of robust studies identified in this review, our findings suggest that to date, workplace interventions have had mixed effects on the dietary and physical activity behaviours of school staff, with most trials reporting statistically non-significant results. No studies were identified that addressed sleep behaviours of educational staff and none were identified that were conducted in childcare settings. There is a need for additional studies to expand the knowledge on health behaviours in childcare staff and interventions to improve those.

Only one of six included studies that targeted diet found an intervention effect for dietary practices (i.e., fruit and vegetable intake) [34]. Previous reviews of health promotion interventions conducted in workplaces more broadly have found moderate improvements in participants’ dietary outcomes (particularly for fruit and vegetable intake) [20,21,40]. Workplace-based interventions most likely to be effective in improving staff diets have facilitated behaviour change by including environmental modifications (e.g., change of cafeteria menus, greater availability of healthy options) as part of comprehensive workplace programs [20]. In contrast, the studies in the current review predominantly targeted behaviour change (i.e., diet) through intrapersonal or social level strategies such as personalised action plans, walking groups. Programs that target the physical environment in future school-based studies are therefore warranted, with researchers also taking into account the unique characteristics of the setting as a whole (e.g., urbanisation, socio-economic disadvantages).

Two of the three studies that targeted physical activity did not find an intervention effect [17,37]. Both of these studies had follow-up periods of 2 years, whereas the third study with a shorter follow-up period (i.e., 4 months) found that participants in the intervention group reported engaging in significantly more leisure-time physical activity compared to those in the control group (based on univariate analyses with results from multivariate analyses approaching statistical significance) [36]. The literature shows that physical activity interventions in the workplace typically have shorter follow-up periods [24], and the maintenance of behaviour change over longer periods may require dedicated strategies that foster habit formation, self-determination, and enhance enjoyment and satisfaction [41,42,43]. A previous review of workplace physical activity interventions found the use of physical activity self-monitoring (i.e., asking participants to keep a record of specified behaviour/s, for example in the form of a diary [44]) was associated with greater intervention effectiveness [30] and should be considered for future studies [22]. Additional work is required to identify the mechanisms by which interventions in educational staff are effective.

In respect to the secondary outcomes of the review, only two studies assessed staff physical health (i.e., BMI, waist-to-hip ratio) and found no effect [17,37]. Previous reviews of workplace interventions focussing specifically on weight loss found such interventions can produce modest weight loss [45,46]; however, these interventions are often relatively intensive, which may not be appropriate for scale. A recent review of obesity prevention initiatives for health care workers found behavioural interventions delivered by trained professionals via phone, or internet were effective in improving weight-related outcomes [47]. Such delivery modes could be considered for use in educational settings. No study examined the efficacy of the intervention on staff’s mental health. Given the high stress levels in the teacher population [48,49] and known associations between mental health and lifestyle behaviours (i.e., healthy diet and sufficient physical activity) [50,51], this is an opportunity for future research.

Teachers who set positive examples of healthy eating and physical activity amongst students and the school community, may influence students’ health and behavioural outcomes positively and can reduce economic costs to schools [18]. However, as only one study reported spill-over effects on student outcomes [17], it remains to be seen what effect such interventions have on student health. Thus, additional evidence is needed to better understand potential spill-over effects from staff to students following intervention at the staff level.

Whilst this review undertook a comprehensive search and utilised robust review methods, a number of limitations are worth considering when interpreting the review findings. Only studies published in English language were included, which may have excluded other efficacious studies. The studies included in this review were conducted in North America, East Asia and South Africa and none were conducted in Europe or Australia. It is possible that the unique cultural characteristics of each region have an influence on the delivery of workplace-based health promotion (i.e., program conceptualisation and delivery). Therefore, it is important to note that the generalisability of findings from this review is constrained by the characteristics of the individual studies as well as the strategies and components used, and the outcomes assessed in these studies.

A number of the characteristics of included studies were consistently assessed as having high risk of bias. Additionally, of concern was the lack of blinding of outcome assessors, which may reduce confidence in individual trial findings. Future workplace-based health promotion studies in educational settings should consider addressing some of the sources of risk of bias identified in this review. For example, research personnel should be blinded to group allocation where possible, and intention-to-treat analyses should be used where appropriate. Moreover, the routine publication of study protocols prior to conducting trials would be desirable, as it provides an opportunity to make sufficient detail about the intervention design and methods available, which is essential for data synthesis, and also shows to what extent an intervention was delivered as planned.

Some of the studies included only small numbers of schools and staff [38,39], which may have reduced the power to detect significant between-group changes. Further, not all of the studies dealt with potential clustering effects (i.e., school-specific characteristics) in an appropriate manner and this may have had an additional impact on estimates of dietary and/or physical activity behaviours. None of the studies specifically recruited individuals with poor dietary and physical activity habits. As a result, there may have been little room for improvement from baseline. This was evident in some of the studies, where large proportions of participants met recommendations for fruit and vegetable intake and engaged in sufficient physical activity before commencing the intervention (e.g., Siegel et al.) [37]. Finally, all of the included studies used self-report measures, the validity of which was not reported for all, which may have increased overall bias in the reported findings.

## 5. Conclusions

Educational settings employ a substantial proportion of the workforce [2]. Thus, these settings have the potential to significantly influence the health behaviours of a large proportion of adults and that of children [3]. Only a small number of studies were identified, and findings cannot be generalised beyond the cultural and organisational contexts and the strategies used in the studies included in this review. Due to moderate risk of bias in the included studies, it is difficult to draw reliable conclusions about the efficacy of these interventions. The paucity of rigorous studies in educational settings identifies a gap where additional evidence is needed and indicates that more rigorous work in this field is warranted.

## Figures and Tables

**Figure 1 ijerph-17-04998-f001:**
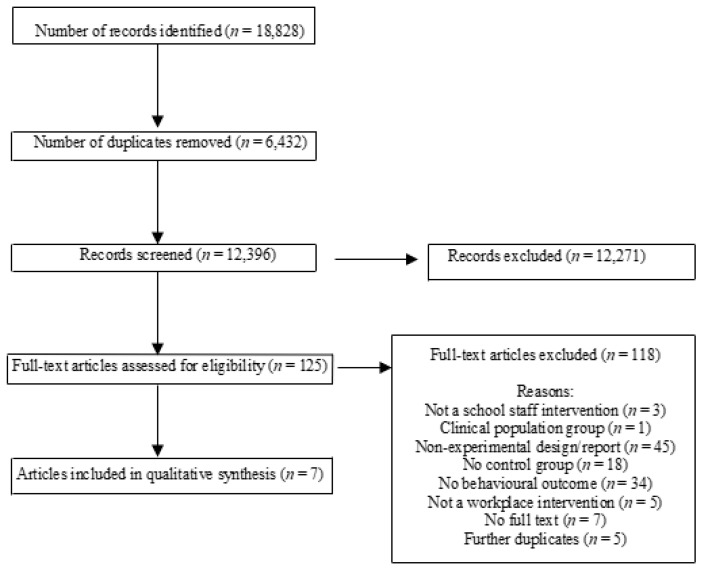
PRISMA diagram illustrating the flow of records in this systematic review.

**Table 1 ijerph-17-04998-t001:** Study characteristics.

Authors (Year)	Intervention Description	Comparator Description	Study Design and Setting (Duration)	Sample Size *	Outcomes **	Statistical Analyses ***	Main Findings
Chen, 2010 [34]	Health promoting schools program targeting teachers’ diet (HP-D)	(1) Health promoting schools’ program not specific to diet (HP-ND) (2) No health promoting schools program control (NHP)	Non-randomised controlled matched group pre-post design in four Taiwanese elementary schools (2 years)	*n* = 283	Self-reported nutrition knowledge (i.e., food classification, nutrient functions, weight control, food safety, recommended daily consumption for children) and dietary behaviours (i.e., reading nutritional information labels, eating breakfast daily, water intake, adhering to ‘five-a-day’, frequency of consuming meat, vegetables, fruits, snacks and sweetened beverages)	Multivariate linear regression models, comparing HP-D to NHP and HP-D to HP-ND, stratified by BMI, weight perception, physical fitness satisfaction, teaching-related status items, adjusted for age, sex and marital status	**Nutrition Knowledge** Between-group differences in nutritional knowledge scores were statistically significant, with teachers in the HP-D group on average reporting higher nutritional knowledge scores relative to NHP teachers (*p* < 0.001). **Dietary Practices** Between-group differences comparing teachers in the HP-D group to those in the NHP group were statistically significant for fruit and vegetable intake (*p* = 0.01), but not for nutrient intake and consumption of fatty foods. No statistically significant group-differences were found between the HP-ND and NHP groups.
Kupolati, 2019 [39]	Multi-component nutrition education program for teachers	No intervention control	Quasi-experimental controlled trial in two South African elementary schools (8 months)	Teachers *n* = 23 Students *n* = 681	Self-reported nutrition knowledge (i.e., current dietary recommendations for children, sources of nutrients, diet-disease relationship, food processing, and food hygiene), nutrition attitude and dietary practices (i.e., personal dietary practices, dietary practices at school, classroom food practices, schoolwide food practices, and practices in food hygiene), students’ nutritional knowledge and attitudes (e.g., nutrient sources and functions, energy, food choices) and dietary practices	Random effects generalised least squares (GLS) regression models	**Nutrition Knowledge** Between-group differences in changes in total nutrition knowledge scores from pre- to post-implementation were statistically significant with the treatment school reporting greater changes in scores compared to the control school (*p* = 0.003). The treatment school had higher mean scores than the control school in all sub-categories of nutrition knowledge. Between-group differences in changes from pre to post however, were non-significant, except for the nutrient sources category (*p* < 0.001). **Nutrition Attitudes** Between-group differences in changes in nutrition attitudes from pre to post were statistically non-significant. **Dietary Practices** Between-group differences in changes in dietary practices from pre to post were statistically non-significant. **Student Outcomes** Between-group differences in changes from pre to post were statistically significant and in favour of the IG for students’ total nutrition knowledge scores (*p* = 0.001) and attitudes (*p* = 0.002), but not for students’ dietary practices.
O’Loughlin, 1996 [36]	Multi-component health screening and counselling program targeting teachers’ physical activity and nutrition	No intervention control	Non-randomised matched control group pre-post design in 16 Canadian elementary schools (4 months)	*n* = 260	Self-reported dietary intake (i.e., frequency of fatty food consumption) and leisure time exercise behaviour (days/week of strenuous, moderate and mild exercise)	Multivariate ANCOVAs, adjusted for baseline, age, language, and self-reported high blood pressure	**Dietary Practices** Between-group differences in changes in fat consumption were statistically non-significant, with both groups showing a slight increase at post. **Physical Activity** The difference in the proportion of teachers who increased their leisure-time exercise behaviour scores comparing participants in the IG (62.1%) to those in the CG (47.3%) was statistically significant (*p* = 0.02) ^a^. The difference in adjusted mean differences for this outcome, however, was statistically non-significant (*p* = 0.05).
Resnicow, 1998 [17]	Multi-component teacher wellness program targeting physical activity and nutrition (Johnson and Johnson Live for Life)	Teachers and students in the control group received the Gimme-5 program (i.e., health education curriculum based on social cognitive theory designed to increase students’ consumption of fruits and vegetables)	Cluster-randomised matched control group design in 32 U.S. elementary schools (2 years)	Teachers Full sample: *n* = 439; Cohort: *n* = 96 Students Full sample: *n* = 3052; Cohort: *n* = 966	Dietary intake via 7-day food diary (to monitor day, time, meal, location, number of servings, food preparation procedures, rating of fat content and high/low fat practices), exercise habits via 7-day exercise diary (to monitor day, level of effort, duration in minutes, which were used to calculate total weekly minutes, METs and energy expenditure in kcals), assessor-measured waist-to-hip ratios, skinfolds and blood pressure, self-reported job satisfaction, perceived organisational climate and student behaviours (as listed above for teachers)	Mixed model repeated-measures ANCOVAs including fixed effects for experimental condition and ethnicity and random effects for school (nested in treatment condition) and individual (nested within school). In addition, student outcomes included an additional fixed effect for gender	**Dietary Practices** Between-group differences in dietary practices were statistically non-significant using both complete and incomplete data. Physical Activity Between-group differences in physical activity were statistically non-significant using both complete and incomplete data. **Physical Health** Between-group differences in all of the specified physiological and anthropometric outcomes were statistically non-significant. **Student Outcomes** Using complete data, no evidence was found that the teacher intervention modified student health behaviours. Using incomplete data (i.e., full sample), however, there was a statistically significant effect on students’ fruit and vegetable preferences (*p* < 0.01) favouring the control group.
Siegel, 2010 [37]	Multi-component obesity prevention program targeting school staff’s physical activity and nutrition	No intervention control	Cluster-randomised controlled trial in 16 U.S. elementary schools (2 years)	*n* = 672 (for analysis of fruit and vegetable intake) *n* = 650 (for analysis of physical activity) *n* = 676 (for analysis of BMI) *n* = 701 (for analysis of waist-to-hip ratio)	Self-reported physical activity (IPAQ-SF), fruit and vegetable consumption (i.e., NCI All-day screener which is a food frequency questionnaire), anthropometric measures (i.e., height, weight to calculate BMI, waist and hip circumference to calculate waist-to-hip ratios) taken by trained study personnel	Linear mixed models including fixed effects for treatment condition and time and random effects for school worksite (nested within treatment condition) as well as individuals (nested within schools), adjusted for age, ethnicity, job classification	**Dietary Practices** Between-group differences in fruit and vegetable consumption were statistically non-significant. **Physical Activity** Between-group differences in physical activity were statistically non-significant. **Adiposity** Participants in the IG on average reported reductions in their BMI (-0.04 kg/m^2^), whereas controls increased their BMI by 0.37 kg/m^2^. This between-group difference was statistically significant (*p* = 0.048). Between-group differences in participants’ waist-to-hip ratio were statistically non-significant.
Wang, 2016 [38]	Multi-component health promoting schools program targeting teachers’ nutrition	No intervention control	Cluster-randomised controlled trial in two Chinese middle schools (6 months)	*n* = 40	Self-reported nutrition knowledge (i.e., nutrient function, nutrient content, food poisoning, nutrition attitudes towards nutrition, healthy dietary habits and food safety (i.e., food expiry dates) and frequency of consumption (i.e., fresh fruits and vegetables, dairy products, breakfast, dessert, fried foods and soft drinks)	Chi-squared tests	**Nutrition Knowledge** Between-group differences were statistically non-significant for nutrition knowledge and awareness. **Nutrition Attitudes** Between-group differences in nutrition attitudes were statistically non-significant. **Dietary Practices** Between-group differences in dietary practices were statistically non-significant.

* N used for analysis; ** only the behavioural and health outcomes relevant to this review are listed. Studies may have assessed a broad range of other (unrelated) outcomes; *** alpha levels for statistical significance were set to 0.05 for all analyses, except for Kupolati et al. [39] (alpha = 0.025) and Resnicow et al. [17] (alpha = 0.001). ^a^ This result was based on univariate analyses and included in this table due to its meaningfulness from a public health perspective. IG = intervention group; CG = control group; SCT = Social Cognitive Theory; MLM = Meaningful Learning Model; IPAQ-SF = International Physical Activity Questionnaire Short Form; NCI = National Cancer Institute.

**Table 2 ijerph-17-04998-t002:** Risk of bias in cluster-randomised controlled trials.

Study	Random Sequence Generation (Selection Bias)			Allocation Concealment (Selection Bias)			Blinding of Participants and Researchers (Performance Bias)			Blinding of Outcome Assessment (Detection Bias)			Incomplete Outcome Data (Attrition Bias)			Selective Reporting (Reporting Bias)			Recruitment Bias			Loss of Clusters			Incorrect Analysis			Contamination			Baseline Imbalance		
Resnicow, 1998 [17]		?			?			–			?			+			?			?			–			+			+			?	
Siegel, 2010 [37]		?			?			–			?			−			?			?			?			+			+			+	
Wang, 2016 [38]		?			?			–			?			+			?			+			+			+			+			+	

*Note.* Low risk of bias (+); unclear risk of bias (?); high risk of bias (−).

**Table 3 ijerph-17-04998-t003:** Risk of bias in non-randomised controlled trials (based on the Newcastle Ottawa Scale).

Criteria			Rating Scale	Chen, 2010 [34]	Kupolati, 2019 [39]	O’Loughlin, 1996 [36]
SELECTION	Representativeness of the intervention cohort	Truly representative of the average school staff member	1			
		Somewhat representative of the average school staff member	1	X		X
		Selected group of patients (e.g., only certain socio-economic groups/areas)	0		X	
		No description of the derivation of the cohort	0			
	Selection of the non-intervention cohort	Drawn from the same community as the intervention cohort	1	X	X	X
		Drawn from a different source	0			
		No description of the derivation of the non-intervention cohort	0			
	Ascertainment of intervention	Secure record (e.g., health care record)	1			X
		Structured interview	1		X	
		Written self-report	0			
		Other/no description	0	X		
COMPARABILITY	Comparability of cohorts on the basis of the design or analysis *	Study controls for age, sex, marital status	1	X	X	X
		Study controls for additional factors (e.g., socio-economic status, education)	1	X		X
		Study does not control for or report factors that reduce comparability of cohorts (i.e., analyses unadjusted)	0			
OUTCOME	Assessment of outcome	Independent blind assessment	1			
		Record linkage	1			
		Self-report	0	X	X	X
		Other/no description	0			
	Was follow up long enough for outcomes to occur	Yes, if median duration of follow-up was ≥3 months	1	X	X	X
		No, if median duration of follow-up was <3 months	0			
	Adequacy of follow up of cohorts	Complete follow up (all subjects accounted for)	1			
		Subjects lost to follow up (i.e., ≤20%) unlikely to introduce bias	1		X	
		Follow up rate <80% and no description of those lost	0			X
		No statement	0	X		
		Overall rating		5	5	6

* a maximal rating of 2 was possible for this item, and the maximal rating for all other items was 1 (out of a possible overall rating of 8); X indicates the criterion was fulfilled.

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
