# Peer review of "The Efficacy of Workplace Interventions on Improving the Dietary, Physical Activity and Sleep Behaviours of School and Childcare Staff: A Systematic Review"

_ijerph, 2020, doi:10.3390/ijerph17144998_

Round 1

Reviewer 1 Report

Dear authors,

thank you very much for this overall very well written and clear structured manuscript. You provide a valuable systematic review that pinpoints the nessessety for more comprehensive and well designed interventions to promote teacher and childcare center staff health. Researchers and practitioners can find helpful information for their furture work in your review.

I have formulated several recommentations and questions in respect to the different chapters of your manuskript. Some are minor hints to improve the readability/understanding, some are more general aspects to be considered.

Abstract:

  • Line 29: Here you only mention controlled trials, but also other designs were eligibly for inclusion.
  • Line 32: Six included articles is not completely correct, as you included seven articles, based on six studies. Please check that against lines 231- 232: Seven papers reporting on six unique studies. 
  • Although you have only about 200 words in the abstract, and already written 185 words, the content regarding the acutal gained results is very sparse: "Multiple outcomes were assessed, and findings were mixed." Please consider to reduce some of the basic information/descriptives. You could for instance shortly mention that you applied the PRISMA Guidelines and skip the information regarding double check of titles, abstract, fullt text etc. That is by now common knowledge when it comes to SR and a basic prerequisite if you conduct a SR: no need to mention that. You coult also delete that sentence "Interventions addressed dietary and physical activity behaviours in combination (n=3) or diet only 34 (n=3); none targeted sleep." as it only provides descriptive information regarding the outcome combinations aimed at.  Use the limited words for more content based information.
  • I furthermore suggest not to start with the sentence "There is limited evidence on the efficacy of workplace-based behaviour change interventions in school and childcare settings." as that is indeed the outcome of your SR. I suggest instead to mention that there is a need for such high quality interventions due to teacher health status.

Introduction:

  • Line 53: what happend since 1981? Please refer to at least one reference regarding that topic from the last years.
  • Your line of arguments is a bit confusing, as you mention that the educational setting has potentional for intervention implemention in lines 56-57 and again in 68-72. Please try to better link the arguments to give the reader a better reading flow.
  • Line 58 ff.: please provide some indicators why educational staff could be potentionally at risk for certain health behaviours. Do they have a lack of time to consume heathy food? One could argue that teacher are not forced to sit long working ours in comparison to other office workers, are therefore more physically active. Not all of your cited references seem to have good comparison between teacher population and the general population.
  • Line 63: The role model of teachers is a very important point. Please consider that in more detail in the discussion and/or future directions.
  • Line 72 ff.: What about to report and appraise the contends of the interventions to determine why they worked or not? That would be somehow neccessary to guide the development of future interventions. As you describe these information in the Supplemantary Table 2, please make stronger usage of these information, at least in the discussion. What could have been key aspects of the effective interventions?

Materials and Methods:

  • Well conducted by the use of recognized guidelines. Please update the current status of your review in Prospero
  • Line 117: please consider to use the term "moderate-to-vigorous-intensity physical activity"
  • Line 136 ff.: you made sligthly changes regarding the searched data bases in comparison to your Prospero Protocol. Please mention why you e.g. did not used HMIC database or Physical Education Index.
    Please add "ProQuest" before "Dissertations and Theses"
  • Line 138 ff.: For transparency reasons, could you please provide more information (e.g. what are relevant terms?) for these searches, analogous to your search strategy applied to the databases that is very well described in the supplementary material S1.
  • Line 191: I think you meant "A third reviewer", as you mentioned two reviewers already in line 189.
  • Line 201 ff.: This introduction sentence is misleasding, as one could think that you used the Cochrane tool to assess the risk of bias for all included studies. This does not make sense, as also non-randomised trials were eligably for inclusion and the reader does not know yet what kind of designs you have included. It becomes clear in lines 209 ff., but is confusing in lines 201 ff.
  • Line 231: Please consider the use of "papers" and "articles", see also in the abstract.
  • Line 234 is a bit confusing, as not the related text, but table one follows. That should be rearranged. 

Table 1:

  • please check the formats within this table. Perhaps a wide format is more appropriate as especially the information in variable "Outcomes" are very long in each study and in the long format that is very unpleasend to read. Please also check the writing behind the bullet points in variable "Main findings". Furthermore, there is no clear cut between e.g. the first two studies in the Outcome variable. That is also the case for other studies. For sure, the wrong formatting is perhaps based on the journal producation at that stage of the rewiew process.
  • Please consider the use of "significant" and "statistically significant" within this table and in general. Because of the multiple usage, it is not clear what has been reported, based on which stastitical methods, in the primary studies. What meaning or "practical significance" do these results have? Based on crosscheck of the six primary studies, most presented results in this manuscript represent at least p-values. Therefore, to use the term "statistical significance" migth make sense for all of these results. However, that practise have been widely critised ( Wasserstein, Ronald L.; Lazar, Nicole A. (2016-04-02). "The ASA's Statement on p-Values: Context, Process, and Purpose". The American Statistician. 70 (2): 129–133. doi:10.1080/00031305.2016.1154108).
  • Please check the publication date for Chen (2009): this one seems to be 2010 instead of 2009.
  • Chen, Main Findings, Nutrition Knowledge: In comporison to what? This is not clear here.
  • Chen (2010): It is unclear, how many groups were uncluded and what kind of treadment they got. For example, under Nutrition Knowledge you mention benefits for teachers in the IG. Under Dietary Practise you mention dietary and non-diatary IG. So which group you refer to under Nutrition Knowledge? Based on a crosscheck, that must be the diatary IG. Please do not leave that to be checked by the reader. One solution could be to include that information under Study Design and Setting or to include more details regarding the groups in Supplemantary File S2. I prefere the first solution. as the reader otherwise need to switch a couple of times between the two documents. From my perspective that is very important for the interpretation of the results. That is a generel problem within the presentation of the results. 
  • Kupolati (2019): Please add student sample size.
  • Please carefully check the sample sizes: e.g. O’Loughlin (1996): you mention n = 209, but that is only the original sample size of the eight intervention groups; you do not report that n = 125 completed the data collection etc. Furthermore, n = 177 were asked to particpate in the CG, of which 135 seem to be included. In comparison to Chen (2010), you repored the full sample size of n = 361; n = 281 were included. Please add these information regarding overall sample size and split group retention rate and/or split group included sample size. That is very important, as e.g. Siegel (2010) differentieted subgroups based on available pre- und post intervention data. From the overall n = 413, only n= 125 seem to have both baseline and post intervention data and where therefore included in specific models.
  • Resnicow (1998), Main Findings, Diatary Practise: "No significant between" = please consider the term/spelling.
  • Resnicow (1998): Here it becomes obvious why more details on sample size are neccessary: you differentiate the main findings for diatary practise into complete data and incomplete data, indicating different results. A cross check with Tab. 2 from the primary studies reveales that the complete data is based on n = 96 observations, the incomplete data (or full samle) is based on n = 436 observations, which is also somehow confusing as it is mentioned that initially n = 316 teachers were eligibly for baseline data collection. The n = 233 you mentioned completed the baseline data collection. So here we have again a mismatch as you sometimes report the sample size of the full sample, and sometimes not. Furthermore, please provide sample size information on student level, as you report results on student level.
  • Resnicow (1998):Why you do not report the significant result on student level based on incomplete data? That is especially interesting as in indicates a benefit towards the CG.
  • Please carefully check the Main Findings with respect to the conducated analysis: e.g. you report that Siegel (2010) adjusted the linear mixed model (LMM) for Adiposity as a DV for age, ethnicity, job classification and clustering effects . However, the covariates age, ethnicity, job classification have been used in all LMMs (DVs: BMI, waist–hip ratio, metabolic equivalent minutes of activity per week, and daily fruit and vegetable consumption) in a second step. From my perspective, the mentioned clustering effect is not a covariate, as the school worksite was included as a random effect.
  • In general, it would be very helpful if you add information regarding the applied statistical analysis for the six primary studies in Table 1. That would make it quite easy for the reader to understand on which statistical methods the results are based on.

Particpants:

  • Line 242 ff: please mention information in student level, as you report results on students level in Kupolati (2019) and
    Resnicow (1998)
  • Line 271: double "of the"

Outcomes:

  • Line 289: Consider italics for "Secondary outsomes" as you use it in "Primary Outcomes"
  • Line 295: Citation for the study/studies is missing. Furthermore, not only one but two studies examined outcomes on student level

Retention and adherence:

  • Line 304 ff: please consider where to mention the citation numer; in comparison to line 303.

Table 2:

  • Please consider the same structure as in table 1, with the year of publication. Anyhow, please consider the use of the related numbers in supercript, to give a clear reference in all tables.

Table 3:

  • The formation and the presented results are somehow confusing. What does "AR" before "Comparability of cohorts..." mean? The meaning of 0 and 1 in the rating scale is unclear. Please consider to provide more information under Note.

Study quality of the included non-randomised controlled trials:

  • Line 325: Why 4? Accorcording to the table it should be 6. And how do you come to these ratings based on the table? When is a category suffieciently fullfilled? What does the "X" mean?

Efficacy of school and childcare staff interventions:

  • Line 334: why childcare staff intervention? By now it is clear that no childcare staff intervention is included.
  • Please consider the presentation of results in terms of the paramers β, CI, p-value etc. Sometimes you provide information for non-signifcant results, sometimes not. Please make it consistantly throughout the manuscript.

Intervention effects on diet:

  • Line 340: Why do you mention the used statistical method here, but not for all analyis. I strongly recommend to provide that information for all six studies, but better in Tab.1 and not here.

Interventions effects on physical activity:

  • Line 367: what about the reported effects of change of behaviour, based on odds ratios and CIs? The results you reported here cannot be found in any table in the primary study!?

Intervention effects on physical health:

  • Line 374 ff: please be precise in reporting the results. Here you mentioned that you report the results for the adjusted LLMs. However, these values are based on the unjusted models. 

Intervention effects on student health behaviours:

  • Line 393: Again, please be consistend in reporting the p-values, CIs etc. Here something seems to be statistically significant, but you do not present any numbers.

Discussion:

  • I suggest to sligtly rearrange the beginning of the discussion. You start with, depending on the perspective, a some defending your your self sentence, like "No one has conducted a review, so that is why we do it". But as there is also critique about the inflationary rising number and the general usefulness of SR, I would not start with this argument. Therefore, I suggest to start with a strong argument that the teacher workforce urgently needs well conducted interventions, as they seem to present unhealthy behaviours etc. Then you can argument that it is unclear in the liteture what kind of interventions are effective (by now, no SR has been conducted; please use SR, as there will be for sure some kind of reviews in general focusing that topic).
  • Line 404: please provide a reference for the prevalence.
  • Line 422: I think with self-monitoring you mean some kind of feedback loops for the participants to show them how active or inactive they are? It would be helpful to provide some information what self-monitoring means in that context and how it could precisely influence the effectivness of future teacher health intervention studies.
  • Line 430: consider capital letter "Internet"
  • Line 445: The use of appropriate statistical methods to account for e.g. hierarchical structures etc. is very important. As you mention here that not all studies dealt with this in an appropiate manner, please provide these information somewhere (see also my comments regarding the presentation of the statistical methods in table 1). Otherwise you leave the reader with this important hint, but she or he cannot judge which of the six studies is, based on the statistical methods used, reliable or not. As you have not specifically assessed and appraised the methodological quality with respect to the applied statistical methods of the included studies, please provide at least some kind of overview/comparison of the respective usefulness of the applied statistics.
  • I strongly recomment to discuss the lack of intervention studies focussing child care centers. What does that mean for the staff, as they seem to have the potential bad health conditions (Hoffmann et al., 2013).
  • Please consider my comment in the Introduction regarding the valuable information in Supplemantary Table 2. How can these information be used to discuss the gained results?
  • I strongly recomment to further disucss the risk of bias results in relation to recommentations how to conduct better and more reliable intervention studies in the future. What are take away messages for researcher and practioners? Can all these criteria to minimize the risk of bias be successfully taken into account in educational settings interventions?

Conclusion:

  • Line 454: Please refer to the term "risk of bias" instead of "methodological quality": https://handbook-5-1.cochrane.org/chapter_8/8_2_2_risk_of_bias_and_quality.htm
  • the critical appraisal of methodological quality has often a strong focus on the statistical methods etc., which can be at high level, although there is still a risk of bias due to different reasons
  •  

References:

  • Please carefully check the conistancy, e.g. Line 515: incomplete reference; different use of journal abbreviations; line 568: incomplete reference; different use of DOIs

Reviewer 2 Report

This is an interesting review, and the results of this review shows clear gab in knowledge related to interventions in educational settings. Perhaps the writers are also developing their own intervention on this field as there seem to have conducted little studies on this field. I have some comments related to review, and I consider that these comments could improve the quality of this review a bit.

  • Abstract:
  • “A secondary aim of the review was to assess if such interventions also influenced students’ health behaviours.” 
This sentence is not in line with aims presented in introduction, please modify it.
  • “‘Multiple outcomes were assessed, and findings were mixed.” This sentence is confusing and raises many comments, which outcomes? How findings were mixed? Could you modify it? Or is it necessary to be included?
  • Introduction:
  • Lines 54- 55: represent valuable settings 
.. I know that you are explaining why the schools and preschools are valuable settings later, but maybe you could modify this section a bit so that explanation for valuable setting is stated already after this sentence. It would be beneficial to include something about this mental health topic in this part as well as it is one of your secondary aims. In the discussion, you talk about the high levels of stress, could you mention something about this in intro as well, and something about how stress effects on health.Overall, you could highlight better why specially the educational settings are important instead of other settings.
  • Your secondary aim is to study if the workplace intervention has secondary aim for students’ health, but you have only one short sentence about this in your introduction. Could you please explain a bit more about it? Why the healthy role modelling is important for children?
  • I consider that the “workplace settings” and specially schools and preschools are the key thing in this review. Thus, I would start with the line 52 the whole introduction, and drop off the lines 42-51 or shorten this part and replace them in the other parts of the intro.
  • I wonder if it is important also to mention that school staff is usually low paid workers (at least in preschools, and often in schools), and how the socioeconomic status can influence in health, and thus justify the importance of focusing in school settings. Do you consider that it is important to mention something about gender – most the workers are usually female, at least in childcare settings?
  • Lines 66-67: could you set an example in which workplaces the interventions are often conducted according to these reviews?
  • After reading lines 68-72, this is true, but I wonder how relevant this is for justifying implementing interventions in educational settings? For instance, do you consider that it is necessary for staff to use the playground equipment to increase their PA levels? After reading it, I understood that you are searching interventions that are mainly/only implemented in educational settings, and aims improving health behaviors in the educational setting so that after intervention the staff would be more active and eat healthily only in the educational settings, but isn’t the focus for improving the overall health behaviors of staff in everyday life (despite the context)? I will have another comment related to this topic also later, but in general, I think it would be necessary somewhere define what you mean by workplace intervention so that it is clear for readers. Do you search interventions that are implemented in educational settings, but aims to improve the health behaviors of staff in general (despite the context) or do you search interventions that are implemented in educational settings, and aims to improve the health behaviors in staff in this setting? If this latter option, so how sleep can be improved in educational settings? Please, see my comment in material and methods-section about this same theme.
  • So, as you may already notice, I suggest you to restructure this introduction so that you justify the importance of educational settings, and then importance of focusing on staff in educational settings so that you justify all your aims (e.g. mental health, stress).    

Materials and methods:

Lines 101-102: shouldn’t be school or childcare staff?

The following sentence: Studies were also included if they primarily targeted student health behaviours (e.g., classroom physical activity), but actively involved staff in the intervention, or engaged staff to deliver the intervention to students and also measured outcomes in school and/or childcare staff.

So, if intervention is delivered in preschools aiming to increase children’s PA, and one of the components in intervention is physical education lessons in preschools, which is delivered by staff, so this kind of study is also included or did I understand this correctly? If so, I am pretty sure that there are then more intervention studies conducted than presented in this review. In addition, what are the measured outcomes? When looking at your figure 1, the excluded articles include e.g. not a school staff intervention or not a workplace intervention. So, I wonder if the aim is to find interventions that mainly focus in workplace settings? If so, maybe you could add it clearly before the lines 101-102, and please rephrase the sentence mentioned above.   

Results

Line: 386, For reader, it would valuable to know if the intervention neither had effect on staff. Or how was it?

Discussion:

  • Line 404: Given the prevalence of inadequate dietary, physical activity and sleep behaviours in teachers, 
I think this needs reference.
  • Line 406: Only one study, please add statement “of many found studies? “ 

Lines 412-413: In contrast, the studies in the current review predominantly targeted behaviour change (i.e., diet) through intrapersonal or social level strategies. Please, set examples for intrapersonal and social level strategies.

Most of the found studies were implemented in different countries and in different regions. So, you could state something about this in discussion as well. There were no studies from Europe or from Australia, for instance. In addition, the educational settings are different in each cultural context, so you could state something about this as well. The results of this review are not generalizable, and there is definitely need to implement more interventions in different countries and in different cultural context. Or is there? The discussion could be strengthened by mentioning why it is still important to focus on educational settings.  

Could you mention some reasons why there are no implemented interventions in preschool? Was this surprising? Or overall, that the number of implemented interventions was that low?  I wonder that it is often assumed that childcare staff is an important role model for children, and there are plenty of theories about this topic, but how well we actually know the actual health behaviors of staff. You could promote somehow in the discussion that this theme is still important, and present the gaps in knowledge clearly.

  • Overall, you are giving many examples what to take into account in future interventions, many of them are good examples. But I keep still considering the option that what if the presented interventions were not successful (statistically) because there was nothing to be improved. For instance, you state in “The study that examined teachers’ job satisfaction and perceived organisational climate reported that no significant group differences were found for these outcomes” what is the workplace productivity was already so high that there was no room for improvement? And how you can be sure that your suggestions would work also in educational settings? Thus, I don’t know how relevant it is to provide list of strategy suggestions for future interventions, as your aim in this review is to find out what is known about the topic,and this should be summarized clearly in the discussion and ponder what is the reason for this result, and provide some suggestions for future studies in general level. For me, the summary of your review is that there are low number of implemented interventions in educational settings that are also conducted in different cultural settings with different intervention strategies focusing on different behaviors and measuring these behaviors with poor methodological quality. Thus, indicating the limited knowledge on field, and needing more research to understand the educational setting itself. I consider that this key message should be highlighted better in discussion. In addition, when considering the future studies, I consider that firstly, there should be conducted more studies to understand the health behaviors of staff in educational settings, secondly, to find out suitable methods used in this setting, and then implement interventions.  
  • Conclusion:
  • Line 451-453: Educational settings employ a substantial proportion of the workforce14 and have the potential to significantly influence the health behaviours of a large proportion of adults and potentially also improve children’s health 

  • Could you restructure and shorten this sentence so that it is easier to understand?

Reviewer 3 Report

Thank you for asking me to review this article.

I have a few comments and suggestions to help the interpretation for readers.

Please confirm the attached file.
